# Understanding Embodied Reference with Touch-Line Transformer

**Yang Li**[1✉]**, Xiaoxue Chen**[1]**, Hao Zhao**[1✉]**,**
**Jiangtao Gong**[1]**, Guyue Zhou**[1]**, Federico Rossano**[2]**, Yixin Zhu**[3✉]
[1] Institute for AI Industry Research, Tsinghua University
[2] Department of Cognitive Science, UCSD [3] Institute for Artificial Intelligence, Peking University
✉ liyang@air.tsinghua.edu.cn, zhaohao@air.tsinghua.com,
  yixin.zhu@pku.edu.cn
https://yang-li-2000.github.io/Touch-Line-Transformer

## Abstract

We study embodied reference understanding, the task of locating referents using embodied gestural signals and language references. Human studies have revealed that, contrary to popular belief, objects referred to or pointed to do not lie on the *elbow-wrist line*, but rather on the so-called *virtual touch line*. Nevertheless, contemporary human pose representations lack the virtual touch line. To tackle this problem, we devise the touch-line Transformer: It takes as input tokenized visual and textual features and simultaneously predicts the referent's bounding box and a touch-line vector. Leveraging this touch-line prior, we further devise a geometric consistency loss that promotes co-linearity between referents and touch lines. Using the touch line as gestural information dramatically improves model performances: Experiments on the *YouRefIt* dataset demonstrate that our method yields a +25.0% accuracy improvement under the 0.75 IoU criterion, hence closing 63.6% of the performance difference between models and humans. Furthermore, we computationally validate prior human studies by demonstrating that computational models more accurately locate referents when employing the *virtual touch line* than when using the *elbow-wrist line*.

## 1 Introduction

Understanding human intents is essential when intelligent robots interact with humans. Nevertheless, most prior work in the modern learning community disregards the multi-modal facet of human-robot communication. Consider the scenario depicted in Figure 1, wherein a person instructs the robot to interact with a chair behind the table. In response, the robot must comprehend what humans are referring to before taking action (*e.g.*, approaching the object and cleaning it). Notably, both **embodied** gesture signals and language **reference** play significant roles. Without the pointing gesture, the robot could not distinguish between the two chairs using the utterance *the chair that is occluded*. Likewise, without the language expression, the robot could not differentiate the chair from other objects in that vicinity (*e.g.*, bags on the table). To address this deficiency, we investigate the embodied refer-

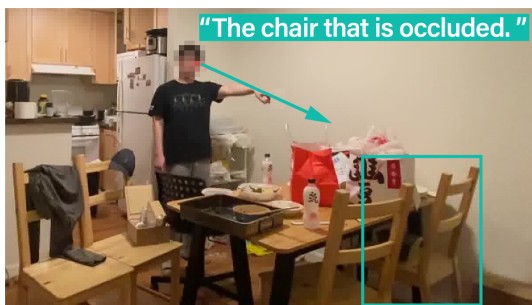

Figure 1: **To accurately locate the referent in complex scenes, both nonverbal and verbal expressions are vital.** Without nonverbal expression (in this case, the pointing gesture), the verbal expression ("the chair") cannot uniquely refer to *the* chair because multiple chairs are present in this context. Conversely, one cannot distinguish the intended referent "the chair" from other nearby objects with only nonverbal expressions.

ence understanding (ERU) task introduced by Chen et al. (2021). This task requires an algorithm to detect the referent (referred object) using (i) an image/video containing nonverbal communication signals and (ii) a sentence as a verbal communication signal.

The **first** fundamental challenge in tackling ERU is the representation of human pose. The *de facto* and prevalent pose representation in modern computer vision is defined by COCO (Lin et al., 2014)— a graph consisting of 17 nodes (keypoints) and 14 edges (keypoint connectivities). Existing models for ERU (Chen et al., 2021) assume pre-extracted COCO-style pose features to be the algorithm inputs. However, we rethink the limitations of the COCO-style pose graph in the context of ERU and uncover a counter-intuitive fact: The referent does not lie on the elbow-wrist line (*i.e.*, the line that links the human elbow and wrist). As shown in Figure 2, this line (in red) does not cross the referred microwave, exhibiting a typical misinterpretation of human pointing (Herbort & Kunde, 2018).

A recent developmental study (O'Madagain et al., 2019) presents compelling evidence supporting the above hypothesis. It studies how humans mentally develop pointing gestures and argues that it is a virtual form of *reaching out to touch*. This new finding challenges conventional psychological views (McGinn, 1981; Kita, 2003) that the pointing gesture is mentally a behavior of using the limb as an arrow. Inheriting the terminology in O'Madagain et al. (2019), we coin the red line in Figure 2 as an elbow-wrist line (EWL) and the yellow line (which connects the eye and the fingertip) as a virtual touch line (VTL). Inspired by this essential observation that VTLs are more accurate than EWLs in embodied reference, we augment the existing

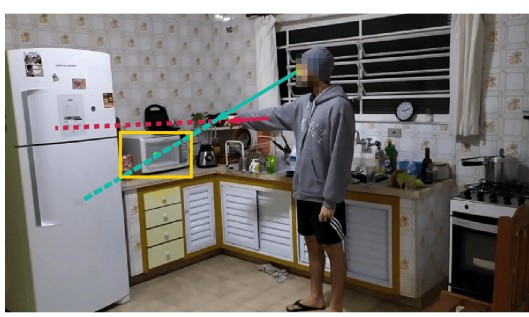

Figure 2: **Virtual touch line (VTL) (in green) vs. elbow-wrist line (EWL) (in red).** VTLs affords a more accurate location of referents than EWLs.

COCO-style pose graph with an edge that connects the eye and the fingertip. As validated by a series of experiments in Section 4, this augmentation significantly improves the performance on *YouRefIt*.

The **second** fundamental issue in tackling ERU is how to jointly model gestural signals and language references. Inspired by the success of multi-modal Transformers (Chen et al., 2020; Li et al., 2020; Tan & Bansal, 2019; Lu et al., 2019; Kamath et al., 2021) in multi-modal tasks (Hudson & Manning, 2019; Antol et al., 2015; Zellers et al., 2019), we devise the Touch-Line Transformer. Our Transformer-based model takes as inputs both visual and natural-language modalities. Our Touch-Line Transformer jointly models gestural signals and language references by simultaneously predicting the touch-line vector and the referent's bounding box. To further help our model utilize gestural signals (*i.e.*, the touch-line vector), we integrate a geometric consistency loss to encourage co-linearity between the touch line and the predicted referent's location, resulting in significant performance improvements.

By leveraging the above two insights, our proposed method achieves a +25.0% accuracy gain under the 0.75 IoU criterion on the *YouRefIt* dataset compared to state-of-the-art methods. Our approach closes 63.6% of the gap between model performance and human performance.

This paper makes four contributions by introducing (i) a novel computational pose representation, VTL, (ii) the Touch-Line Transformer that jointly models nonverbal gestural signals and verbal references, (iii) a computational model leveraging the concept of touch line by a novel geometric consistency loss that improves the co-linearity between the touch line and the predicted object, and (iv) a new state-of-the-art performance on ERU, exceeding the 0.75 IoU threshold by +25.0%.

## 2 RELATED WORK

**Misinterpretation of pointing gestures**    Pointing enables observers and pointers to direct visual attention and establish references in communication. Recent research reveals, surprisingly, that observers make systematic errors (Herbort & Kunde, 2016): While pointers produce gestures using VTLs, observers interpret pointing gestures using the "arm–finger" line. O'Madagain et al. (2019) founded the VTL mechanism: Pointing gestures orient toward their targets as if the pointers were to touch them. In neuroscience, gaze effects occur for tasks that require gaze alignment with finger pointing (Bédard et al., 2008). The preceding evidence demonstrates that eye position and gaze direction are crucial for understanding pointing. Critically, Herbort & Kunde (2018) verify that directing human observers to extrapolate the touch-line vector reduces the systematic misinterpretation during human-human communication. Inspired by these discoveries, we incorporate the touch-line vector to enhance the performance of pointing gesture interpretation.

**Detection of gaze targets**  The gazing gesture itself can refer to particular objects in the absence of language expressions. The detection of the gaze targets is the localization of the objects people gaze at. Recasens et al. (2015) and Chong et al. (2018) introduce two-stage methods that build saliency maps and estimate gaze directions before combining them to predict gaze targets. Tu et al. (2022) devise a one-stage end-to-end approach for simultaneously locating human heads and gaze targets for all individuals in an image. Fang et al. (2021) develop a three-stage method that predicts first the 2D and 3D gaze direction, then the field of views and depth ranges, and finally the gaze target. Zhao et al. (2020) assume that humans find gaze targets from salient objects on the sight line, designing a model to validate this hypothesis in images and videos. Li et al. (2021) extend the detection of gaze target to 360-degree images. Chong et al. (2020) and Recasens et al. (2017) extend the detection of gaze target into videos, wherein the gazed object may not be in the same frame as the gazing gesture.

**Referring expression comprehension**  Language expressions can also refer to particular objects. Referring expression comprehension is to locate referents in an image using only language expressions. Multiple datasets (*e.g.*, RefCOCO (Yu et al., 2016), RefCOCO+ (Yu et al., 2016), RefCOCO-g (Mao et al., 2016), and Guesswhat (De Vries et al., 2017)) serve as the benchmark. To encode images and sentences, Du et al. (2022) develop a Transformer-based framework and use the language to guide the extraction of discriminative visual features. Rohrbach et al. (2016) recognize the bidirectional nature between language expressions and objects in images, obviating the need for bounding box annotations. Yu et al. (2018) devise a modular network, wherein three modules attend to subjects, their location in images, and their relationships with nearby objects, respectively. While the above efforts locate referents using language expressions, they rarely use human-like nonverbal gestures.

**Language-conditioned imitation learning**  Robots must identify target objects when completing goal-conditioned tasks (Stepputtis et al., 2020). Counterintuitively, Stepputtis et al. (2020) argue against describing the target object using (one-hot) vectors due to inflexibility to support continued learning in deployed robots. Instead, Stepputtis et al. (2020) employ natural language to refer to target objects in goal-conditioned manipulation tasks. Specifically, leveraging a semantic module for understanding what the referents referred to by language expressions, their architecture helps robots locate and attend to the target object when the natural language expression unambiguously refers to an object. However, this method struggles to resolve ambiguities (*e.g.*, in cases with pointing gestures). Recent work on language-conditioned imitation learning (Stepputtis et al., 2020; Lynch & Sermanet, 2021; Mees et al., 2022) may therefore benefit from our VTL by more accurately locating the target object, especially when the natural language alone is ambiguous.

## 3  METHOD

### 3.1  NETWORK ARCHITECTURE

Our framework, illustrated in Figure 3, consists of a multi-modal encoder, a Transformer decoder, and prediction heads. We describe each component in detail below.

**Multimodal encoder**  We generate a visual embedding vector by extracting visual features from input images with a pre-trained ResNet (He et al., 2016) backbone, flattening them, and then adding them to a set of position embeddings (Parmar et al., 2018; Bello et al., 2019). Meanwhile, we generate a textural embedding vector from input texts using a pre-trained BERT (Liu et al., 2019). After obtaining visual and textural embedding vectors, we concatenate and feed them into a Transformer encoder to learn multi-modal representations.

**Transformer decoder**  We feed the above multi-modal representations to our Transformer decoder, which additionally takes as input a set of learnable object queries and gestural key point queries. With multi-modal representations and input queries, our Transformer decoder (pre-trained MDETR (Kamath et al., 2021)) generates object output embeddings and gestural output embeddings.

**Prediction heads**  Object and gestural output embeddings from the Transformer decoder are the inputs for our prediction heads (MLPs), which predict bounding boxes (for referents) and gestural key points. We keep one object bounding box and one pair of gestural key points with the highest scores as the final prediction. Specifically, we define the score of a bounding box as 1 minus the non-object class column of the softmax of the predicted object logits. For a pair of gestural key points, the score is the is-a-VTL/EWL column of the predicted arm logits' softmax.

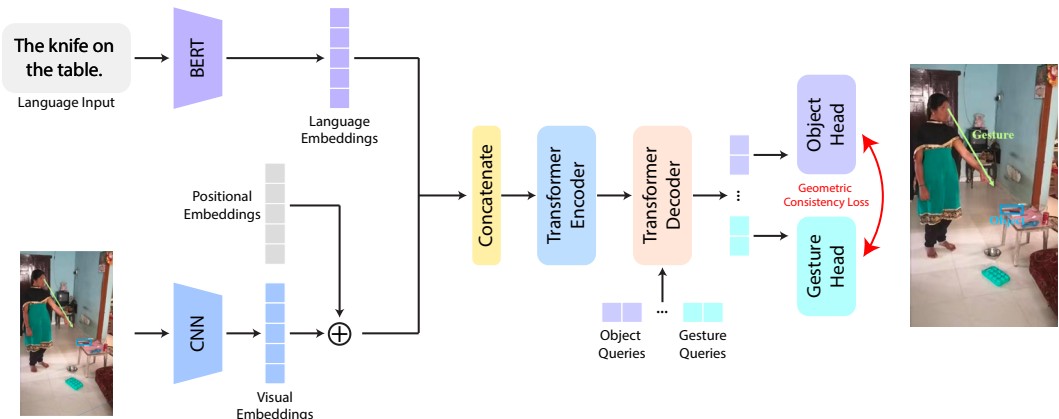

Figure 3: **Overall network architecture.** Language and visual inputs are first encoded by the text encoder and visual encoder to obtain language and visual embeddings, respectively. Next, these embeddings are concatenated and fed into the Transformer encoder to learn multimodal representations. The Transformer decoder and prediction heads output the predicted bounding box and VTL/EWL. A geometric consistency loss is integrated to encourage the use of gestural signals.

## 3.2 EXPLICIT LEARNING OF NONVERBAL GESTURAL SIGNALS

**Measure co-lineaerity**  Ideally, a referent should have a high co-linearity with the VTL. We measure this co-linearity using cosine similarity:

$$cos\_sim = \text{CosineSimilarity}[(x_f - x_e, y_f - y_e),\ (x_o - x_e, y_o - y_e)],  \tag{1}$$

where $(x_f, y_f)$, $(x_e, y_e)$, and $(x_o, y_o)$ are the x-y coordinates of the fingertip, the eye, and the center of the referent bounding box, respectively.

**Encourage co-lineaerity**  We encourage our model to predict referent bounding boxes that are highly co-linear with VTLs using a referent alignment loss, which is a geometric consistency loss:

$$L_{align} = \text{ReLU}(cos\_sim_{gt} - cos\_sim_{pred}).  \tag{2}$$

We compute $cos\_sim_{gt}$ using the ground-truth referent boxes and compute $cos\_sim_{pred}$ using the predicted referent boxes. Ground-truth VTLs are used for both $cos\_sim_{gt}$ and $cos\_sim_{pred}$. Of note, the design of the offset $cos\_sim_{gt}$ ensures the predicted object is as co-linear as the ground-truth object to the VTL. In other words, the loss in Equation (2) is minimized to zero when the predicted object box is as co-linear as the ground-truth one to the VTL.

**Modify for EWLs**  In experiments using EWLs instead of VTLs, we replace the fingertip $(x_f, y_f)$ and the eye $(x_e, y_e)$ in Equation (1) using the wrist $(x_w, y_w)$ and the elbow $(x_l, y_l)$, respectively.

## 3.3 IMPLICIT LEARNING OF NONVERBAL GESTURAL SIGNALS

We eliminate the postural signals (by removing humans) from images to investigate whether our model can implicitly learn to utilize nonverbal gestural signals without being explicitly asked to learn. Please refer to Appendix A for details of our procedure.

## 3.4 ADDITIONAL LOSSES

We define the total loss during training as

$$\mathcal{L}_{\text{total}} = \lambda_1 \mathcal{L}_{\text{box}} + \lambda_2 \mathcal{L}_{\text{gesture}} + \lambda_3 \mathcal{L}_{\text{align}} + \lambda_4 \mathcal{L}_{\text{token}} + \lambda_5 \mathcal{L}_{\text{contrastive}},  \tag{3}$$

where $\lambda_i$s are the weights of various losses, $\mathcal{L}_{\text{box}}$ is the weighted sum of the L1 and GIoU losses for predicted referent bounding boxes, and $\mathcal{L}_{\text{gesture}}$ is the L1 loss for the predicted VTLs or EWLs. The soft token loss $\mathcal{L}_{\text{token}}$ and the contrastive loss $\mathcal{L}_{\text{contrastive}}$ follow those in Kamath et al. (2021); these two losses help our model align visual and textural signals.

# 4 EXPERIMENTS

## 4.1 DATASET, EVALUATION METRIC, AND IMPLEMENTATION DETAILS

We use the *YouRefIt* dataset (Chen et al., 2021) with $2,950$ training instances and $1,245$ test instances. *YouRefIt* has two portions: videos and images; we only use the image portion. The inputs are images *without* temporal information. Each instance contains an image, a sentence, and the location of the referent. We provide additional annotations of VTLs and EWLs for the *YouRefIt* dataset.

For fair comparisons with prior models, we follow Chen et al. (2021) and report precision under three different IoU thresholds: 0.25, 0.50, and 0.75. A prediction is correct under a threshold if its IoU with the ground-truth box is greater than that threshold. Additionally, we choose our best models by the precision under the GIoU (Rezatofighi et al., 2019) threshold of 0.75. Compared with IoU, GIoU is an improved indicator of the model's ability to locate objects accurately. We report the results using GIoU thresholds in Section 4.5.

During training, we use the Adam variant, AMSGrad (Reddi et al., 2018), and train our models for 200 epochs. We use the AMSGrad variant because we observe a slow convergence of the standard Adam optimizer in experiments. We set the learning rate to 5e-5 except for the text encoder, whose learning rate is 1e-4. We do not perform learning rate drops because we rarely observe demonstrable performance improvements after dropping them. We use NVIDIA A100 GPUs. The sum of the batch sizes on all graphic cards is 56. Augmentations follow those in Kamath et al. (2021). The total number of queries is 20; 15 for objects, and 5 for gestural key points.

## 4.2 COMPARISON WITH STATE-OF-THE-ART METHODS

Our approach outperforms prior state-of-the-art methods by 16.4%, 23.0%, and 25.0% under the IoU threshold of 0.25, 0.50, and 0.75, respectively; see Table 1. Specifically, our model performs better than visual grounding methods (Yang et al., 2019; 2020), which do not explicitly utilize nonverbal gestural signals. Our approach also performs better than the method proposed in *YouRefIt* (Chen et al., 2021), which did not leverage the touch line or the Transformer models for multi-modal tasks.

Table 1: **Comparison with the state-of-the-art methods.**

|  | IoU=.25 | IoU=.50 | IoU=.75 |
|---|---|---|---|
| FAOA (Yang et al., 2019) | 44.5 | 30.4 | 8.5 |
| ReSC (Yang et al., 2020) | 49.2 | 34.9 | 10.5 |
| YouRefIT PAF-only (Chen et al., 2021) | 52.6 | 37.6 | 12.7 |
| YouRefIt Full (Chen et al., 2021) | 54.7 | 40.5 | 14.0 |
| Ours (Inpainting) | 59.1 (+4.4) | 51.3 (+10.8) | 32.4 (+18.4) |
| Ours (No explicit gestural key points) | 64.9 (+10.2) | 57.4 (+16.9) | 37.2 (+23.2) |
| Ours (EWL) | 69.5 (+14.8) | 60.7 (+20.2) | 35.5 (+21.5) |
| Ours (VTL) | 71.1 (+16.4) | 63.5 (+23.0) | 39.0 (+25.0) |
| Human | 94.2 | 85.8 | 53.3 |

## 4.3 EXPLICITLY LEARNED NONVERBAL SIGNALS

We report performances of models that explicitly predict either the VTLs or the EWLs.

**Results**  Overall, the model trained to predict the VTLs performs better than the one trained to predict the EWLs under all three IoU thresholds; see details in Table 2. The model trained to explicitly predict the EWLs performs even worse than the one not trained to explicitly predict any gestural signals under the IoU threshold of 0.75.

Table 2: **Effects of learning two different types of postural key points.**

| IoU | None | EWL | VTL |
|---|---|---|---|
| 0.25 | 64.9 | 69.5 (+4.6) | 71.1 (+6.2) |
| 0.50 | 57.4 | 60.7 (+3.3) | 63.5 (+6.1) |
| 0.75 | 37.2 | 35.5 (-1.7) | 39.0 (+1.8) |

**Analysis**  Under the IoU threshold of 0.75, the unreliability of the arm's orientation and the stringent precision requirement can partly explain the worse performance of the model with explicit EWL predictions. As we observed in Figure 4, the EWLs are unreliable for predicting objects' locations simply because they frequently do not pass through the referents. This mismatch prevents the model

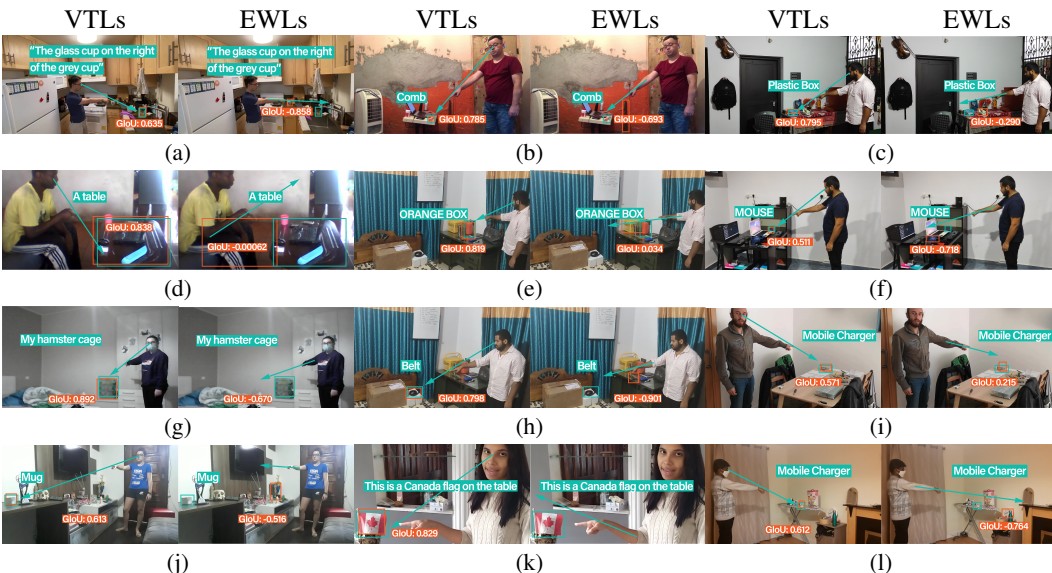

Figure 4: **The model explicitly trained to predict VTLs more accurately locates referents than the one trained to predict EWLs.** We draw green arrows (different from the predictions) to illustrate eyes and fingertips more accurately indicate object locations. (Green sentence: natural language inputs; Green box: ground-truth referent location; Red box: predicted referent location; Red numbers near predicted box: GIoU between the predicted box and the ground-truth box.)

from correctly determining the referents' locations using the EWLs' orientations. For instance, in Figure 4a (right), the EWL fails to pass the glass cup in the yellow box; it passes a ceramic cup instead. The model struggles to identify the referent correctly with contradictory nonverbal (the ceramic cup) and verbal (the glass cup) signals.

In contrast, VTLs partly explains the model's improved performance. For example, in Figure 4b, the VTL passes the comb in the green box while the EWL fails. Similarly, in other subfigures in Figure 4, VTLs passes the referent while EWLs do not, or VTLs are closer to the referents' box centers than EWL are. The higher consistency between verbal and nonverbal signals increases the chance of successfully identifying the referents.

Under the lower IoU thresholds of 0.25 and 0.50, the rough orientations provided by the EWLs and the more lenient precision requirements can partly explain the improved model performances. Specifically, the rough orientations provided by the EWLs might help the model eliminate objects that significantly deviate from this direction. Hence, the model can choose from fewer objects, leading to a higher probability of correctly locating referents.

## 4.4 IMPLICITLY LEARNED NONVERBAL SIGNALS

We investigate whether our model can implicitly learn nonverbal gestural signals without being explicitly asked to predict the VTLs or the EWLs.

To investigate this hypothesis, we conducted two sets of experiments: original images and inpainted images (human removed). In either case, we do not provide our annotated gestural key points (eyes, fingertips, elbows, and wrists) to our model or ask our model to predict any of these gestural key points.

Table 3: **Effects of implicitly learned nonverbal signals.**

| IoU | Original | Inpainting |
|-----|----------|------------|
| 0.25 | 64.9 | 59.1 (-5.8) |
| 0.50 | 57.4 | 51.3 (-6.1) |
| 0.75 | 37.2 | 32.4 (-4.8) |

The set of experiments using original images without providing or predicting gestural key points corresponds to the "Original" column in Table 3, the "No explicit gestural key points" column in Table 1, the "No exp. ges. k. poi." column in Table 7, and the "None" column in Table 2.

Our model performs much worse when nonverbal gestural signals are absent. Specifically, removing humans from input images results in 5.8%, 6.1%, and 4.8% performance drop under the IoU threshold of 0.25, 0.50, and 0.75, respectively; see details in Table 3.

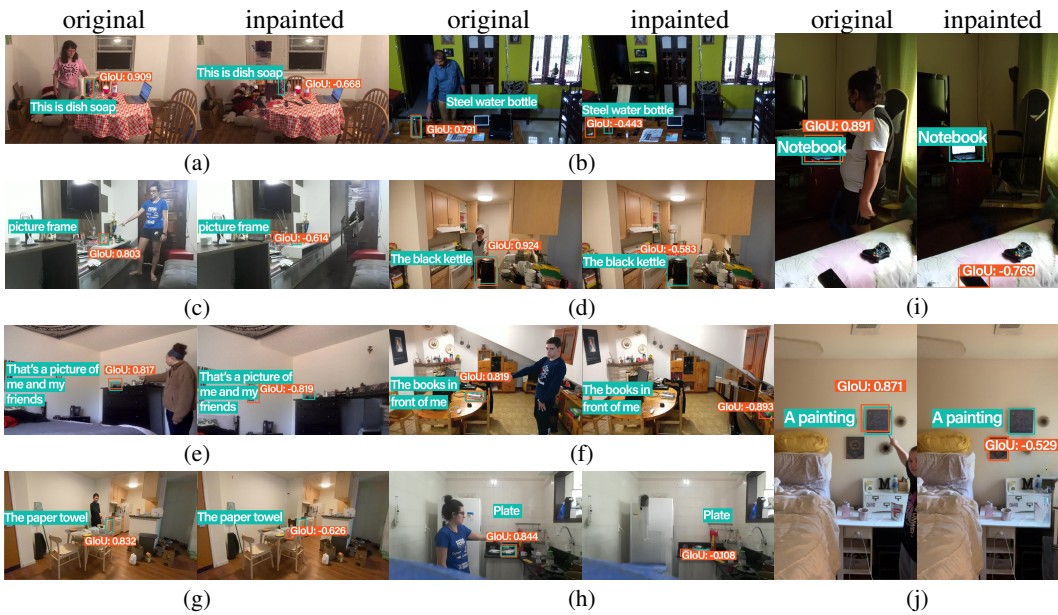

Figure 5: **Training with inpainted images leads to a performance drop.** Removing human contexts increase the difficulties in determining which object is referred to by the natural language expression (a, b, c, i) or even makes it impossible (d, e, f, g, h, j). We call it impossible to determine the referent using natural language input when multiple objects in the image satisfy the descriptions in the natural language input sentence. (Green sentence: natural language input; Green box: ground-truth referent location; Red box: predicted referent location; Red numbers near boxes: GIoU between the predicted box and ground-truth box.)

These performance drops indicate that the model lacks some helpful information when only inpainted images are given. Specifically, removing humans in the scenes eliminates useful communicative signals, including pointing, poses, and the gaze direction (Sebanz et al., 2006). The lack of human context introduces additional ambiguities in identifying the referents, especially when the natural language input alone cannot uniquely refer to an object in the scene.

For example, in Figure 5f, given the natural language input "the books in front of me," determining which piles of books this person refers to is difficult: one on the table and one in the scene's lower right corner, thus in need of nonverbal gestural signals, such as pointing. Similarly, in Figure 5e, we cannot distinguish which picture frame the person refers to without considering the pointing gesture.

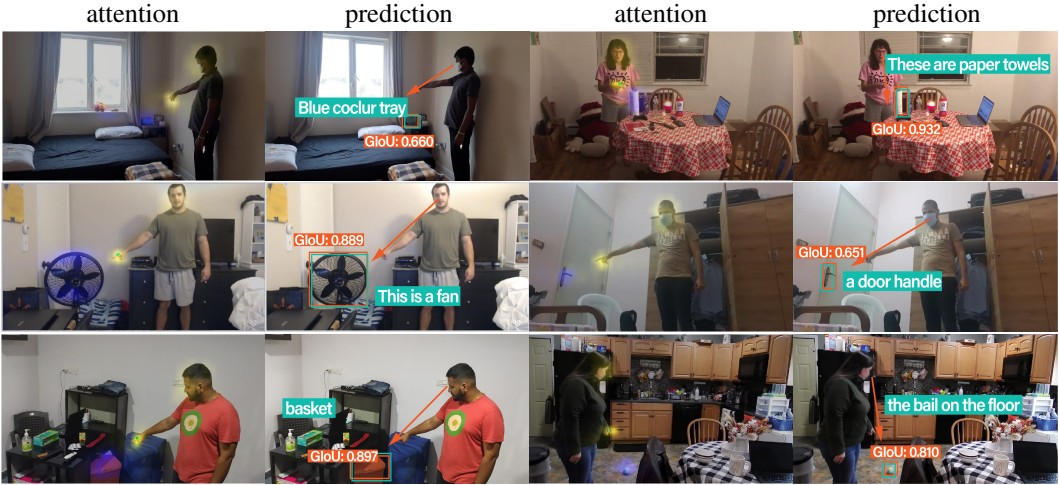

Figure 6: **Attention weights visualizations.** Attention visualization of models trained to learn VTLs explicitly. (Blue: attention from object tokens; Yellow: attention from gestural key point tokens; Green sentence: natural language input; Green box: ground-truth referent location; Red box: predicted referent location; Red numbers near boxes: GIoU between the predicted box and ground-truth box.)

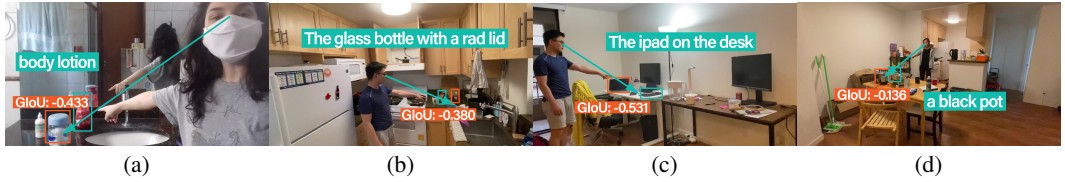

(a)       (b)       (c)       (d)

Figure 7: **Examples of failure cases.** Green sentence: natural language input; Green box: ground-truth referent location; Red box: predicted referent location; Red numbers near boxes: GIoU between the predicted box and ground-truth box.

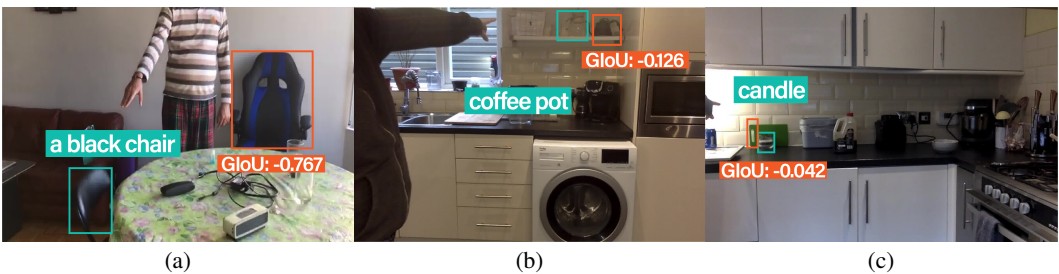

(a)       (b)       (c)

Figure 8: **Rare cases where the human heads are invisible.**

## 4.5 ADDITIONAL RESULTS

**Attention weights visualizations** We visualize the attention weights of our model trained with VTL (Figure 6). We use yellow to visualize the attention weights of matched gesture keypoint queries and blue for matched object queries. Visualizations show that the attention of object queries can attend to the regions of target objects, whereas the attention of keypoint queries primarily focuses on humans' heads and hands. These results indicate that our model successfully learns gestural features that boost performance.

**Failure cases** While the VTL effectively helps the model leverage the nonverbal gestural signals, the performance of our model still could not surpass 74% human performance. The gap between the model and human performance can be partly attributed to the model's inability to distinguish the subtle differences among similar objects. In other words, while explicitly predicting the VTLs helps our model utilize gestural signals to determine the object's direction, it does not help the model recognize the minor differences among multiple objects in the same direction.

For example, the VTL passes two bottles in Figure 7a. While the VTL indeed helps narrow down the options to these two bottles, it does not help our model recognize which one is the body lotion. Similarly, in Figure 7b, the VTL indicates two bottles—one with a red lid and the other with a yellow lid. However, it does not help the model distinguish the subtle differences between lid colors.

Additionally, in very rare cases (Figure 8), failure to detect the human head in the image prevents the application of our VTL.

**Computed cosine similarities** The computed cosine similarity is much higher when using the VTLs (Table 4), indicating the VTLs are more co-linear to the referents. This result verifies our hypothesis that the VTLs are more reliable in referring to the objects' locations.

**Referent alignment loss** An ablation study (Table 5) shows that our reference alignment loss plays a significant role in leveraging nonverbal gestural signals.

Table 4: **Computed cosine similarities.** (tgt: computed using ground-truth box centers; pred: computed using predicted box centers.)

| gesture | cos sim tgt | cos sim pred |
|---------|-------------|--------------|
| EWL     | 0.9580      | 0.9579       |
| VTL     | 0.9901      | 0.9878       |

Table 5: **Effects of the referent alignment loss** $\mathcal{L}_{\text{align}}$**.** Removing the reference alignment loss leads to performance drops in locating referents.

| gesture | ref. align. loss | IoU=.25 | IoU=.50 | IoU=.75 | cos sim pred |
|---------|------------------|---------|---------|---------|--------------|
| VTL     | True             | 71.1    | 63.5    | 39.0    | 0.9878       |
| VTL     | False            | 67.1    | 59.0    | 36.4    | 0.9815       |
|         |                  | (-4.0)  | (-4.5)  | (-2.6)  | (-0.0063)    |

**Effects of object sizes** We evaluate the model performance w.r.t. the object detection of different object sizes (Table 6). We define small (S), medium (M), and large (L) objects following the size thresholds in Chen et al. (2021). Compared to the relatively consistent human performance, deep learning models' performances are significantly lower when detecting diminutive objects than when detecting more prominent objects, especially under the less stringent IoU thresholds of 0.25 and 0.50. The degraded performance indicates that future artificial intelligence models need to improve the performance of small object detection.

Table 6: **Model performances w.r.t. the detection of different object sizes.** Numbers in red and blue denote the highest and the second highest performance, respectively.

| IoUs | 0.25 | | | | 0.50 | | | | 0.75 | | | |
|---|---|---|---|---|---|---|---|---|---|---|---|---|
| Object Sizes | All | S | M | L | All | S | M | L | All | S | M | L |
| FAOA (Yang et al., 2019) | 44.5 | 30.6 | 48.6 | 54.1 | 30.4 | 15.8 | 36.2 | 39.3 | 8.5 | 1.4 | 9.6 | 14.4 |
| ReSC (Yang et al., 2020) | 49.2 | 32.3 | 54.7 | 60.1 | 34.9 | 14.1 | 42.5 | 47.7 | 10.5 | 0.2 | 10.6 | 20.1 |
| YourefIt PAF (Chen et al., 2021) | 52.6 | 35.9 | 60.5 | 61.4 | 37.6 | 14.6 | 49.1 | 49.1 | 12.7 | 1.0 | 16.5 | 20.5 |
| YouRefIt Full (Chen et al., 2021) | 54.7 | 38.5 | 64.1 | 61.6 | 40.5 | 16.3 | 54.4 | 51.1 | 14.0 | 1.2 | 17.2 | 23.3 |
| Ours (Inpainting) | 59.0 | 41.3 | 59.3 | 75.8 | 51.3 | 32.1 | 54.6 | 66.7 | 32.4 | 9.4 | 33.3 | 53.4 |
| Ours (No explicit gestural key points) | 64.9 | 49.6 | 67.9 | 76.7 | 57.4 | 40.8 | 62.1 | 69.0 | 37.2 | 14.4 | 39.7 | 56.7 |
| Ours (EWL) | 69.5 | 56.6 | 71.7 | 80.0 | 60.7 | 44.4 | 66.2 | 71.2 | 35.5 | 11.8 | 38.9 | 55.0 |
| Ours (VTL) | 71.1 | 55.9 | 75.5 | 81.7 | 63.5 | 47.0 | 70.2 | 73.1 | 39.0 | 13.4 | 45.2 | 57.8 |
| Human (Chen et al., 2021) | 94.2 | 93.7 | 92.3 | 96.3 | 85.8 | 81.0 | 86.7 | 89.4 | 53.3 | 33.9 | 55.9 | 68.1 |

**Precision computed using GIoU thresholds** We provide the performance of our models when evaluated using GIoU instead of IoU thresholds; see Table 7.

Table 7: **Model performances when evaluated using GIoU thresholds.**

| | IoU=.25 | IoU=.50 | IoU=.75 |
|---|---|---|---|
| Ours (Inpainting) | 57.9 | 50.9 | 31.4 |
| Ours (No exp. ges. k. poi.) | 63.7 | 56.5 | 36.2 |
| Ours (EWL) | 67.9 | 59.7 | 34.8 |
| Ours (VTL) | 70.0 | 62.5 | 38.2 |

**Language inputs with more ambiguities** We investigate the performance of our model when using language inputs with more ambiguities by changing each input sentence to the word "object." Our results (Table 8) show that language inputs with more ambiguities impair the performance of our models severely.

Table 8: **Model performances when using language inputs with more ambiguities.**

| | More ambiguous | Re-training | IoU=.25 | IoU=.50 | IoU=.75 |
|---|---|---|---|---|---|
| Ours (No Exp. Ges. K. Poi.) | True | False | 19.5 | 16.0 | 7.8 |
| Ours (VTL) | True | False | 31.4 | 25.3 | 13.1 |
| Ours (No Exp. Ges. K. Poi.) | True | True | 49.2 | 42.1 | 23.4 |
| Ours (VTL) | True | True | 49.4 | 41.7 | 23.9 |
| Ours (No Exp. Ges. K. Poi.) | False | False | 64.9 | 57.4 | 37.2 |
| Ours (VTL) | False | False | 71.1 | 63.5 | 39.0 |

## 5 Conclusion and limitations

We presented an effective approach, Touch-Line Transformer, to utilize the simple but effective VTLs to improve an artificial agent's ability to locate referents referred to by humans in the wild. Our approach is inspired by recent findings in human studies on the touch-line hypothesis, which further revealed that people frequently misinterpret other people's referring expressions. Our proposed architecture, combined with the proposed VTL, significantly reduced the gap between model performance and human performance.

Some limitations exist in our work. First, resizing operations before and after inpainting might influence model performance. Next, we primarily study the eyes' location and the upper limbs' orientation regarding nonverbal signals, leaving the study of other types of nonverbal signals to future works, such as gazing direction, the direction of the finger, and the orientation of the lower limbs.

ACKNOWLEDGMENTS

The authors would like to thank Miss Chen Zhen (BIGAI) for making the nice figures and three anonymous reviews for constructive feedback. This work is supported in part by the National Key R&D Program of China (2022ZD0114900), the Beijing Nova Program, and Beijing Didi Chuxing Technology Co., Ltd.

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

# A   INPAINTING

We hypothesize that our model may learn to use postural signals without being explicitly asked to. To test our hypothesis, we remove postural signals (by removing humans) from the images mainly using the Mask R-CNN (Massa & Girshick, 2018) and MAT (Li et al., 2022). We investigate whether our model performs worse when trained using images without gestural signals.

We hypothesize that Transformer models may learn to use postural signals without being explicitly asked to learn these types of signals. To test our hypothesis, we conduct two groups of experiments: the inpainting group and the control group. In the inpainting group, we remove postural signals in the input image. In the control group, we do not modify the input image.

In the inpainting group, we modify input images. Specifically, we remove postural signals from input images by removing humans and filling in missing portions of the image (initially occupied by humans) using MAT (Li et al., 2022). Specifically, we first use Mask R-CNN X-101 (Massa & Girshick, 2018) to produce human masks. After that, we expand the human masks created by the mask rcnn to both the left and right sides to cover the edge of humans entirely. We ensure that the expanded mask never encroaches on regions occupied by the ground truth bounding box for the referent. After that, we feed the expanded masks into MAT. With input masks, MAT removes the regions covered by the masks and fills these regions. Examples of masks, expanded masks, and inpaintings are in Figure A1.

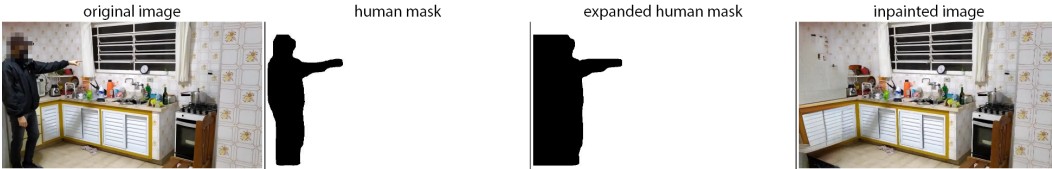

Figure A1: **Illustration of the inpainting process.** We remove gestural signals from input images before feeding images into our model to study the effects of implicitly learned postural signals.

We remove gestural signals (through inpainting) from images when studying our model's ability to learn these signals implicitly. Before generating inpaintings, we expand the human mask to both sides by 50 pixels. We reshape masks and images to $512 \times 512$ before feeding them into the MAT model because the checkpoint produced by MAT only works for inputs of size $512 \times 512$. Outputs of the MAT model are reshaped to their original sizes before feeding into our model. We observe that, for a tiny number of images, human masks cannot be generated by F-RCNNs. In these sporadic cases, we use the original image instead.

# B   ADDITIONAL LOSSES

## B.1   LOSSES FOR PREDICTED REFERENT BOUNDING BOXES

We use a weighted sum of L1 and GIoU losses for predicted bounding boxes. Each bounding box $B = (x, y, w, h)$ is represented using the x and y coordinate of the box center (x and y), width (w), and height (h). We denote the predicted box as $B_p$ and its ground truth as $B_t$.

**L1 Loss**    For each pair of predicted box $B_{p_i}$ and ground truth box $B_{t_i}$, the L1 loss is:

$$L_{L1_i} = |B_{p_i} - B_{t_i}| = |x_{p_i} - x_{t_i}| + |y_{p_i} - y_{t_i}| + |w_{p_i} - w_{t_i}| + |h_{p_i} - h_{t_i}|. \tag{A1}$$

Each time, there are n pairs of predicted and ground truth boxes. The total L1 loss for all pairs is:

$$L_{L1} = \frac{1}{n} \sum_{i=1}^{n} L_{L1_i}. \tag{A2}$$

**GIoU loss**    Before computing the GIoU Loss (Rezatofighi et al., 2019), each box $B = (x, y, w, h)$ is transformed to $\bar{B} = (x_{min}, y_{min}, x_{max}, y_{max})$, where $x_{min} = x - \frac{w}{2}, x_{max} = x + \frac{w}{2}, y_{min} = y - \frac{h}{2}, y_{max} = y + \frac{h}{2}$.

The area of predicted box $\bar{B}_p$ and ground truth box $\bar{B}_t$ are computed as:

$$
\begin{aligned}
Area_{p_i} &= (x_{p_{max_i}} - x_{p_{min_i}}) \times (y_{p_{max_i}} - x_{y_{min_i}}) \\
Area_{t_i} &= (x_{t_{max_i}} - x_{t_{min_i}}) \times (y_{t_{max_i}} - x_{t_{min_i}}).
\end{aligned}
\tag{A3}
$$

The IoU and Union of $\bar{B}_p$ and $\bar{B}_t$ are computed as:

$$
\begin{aligned}
x_{left_i} &= \max(x_{p_{min_i}}, x_{t_{min_i}}) \\
y_{top_i} &= \max(y_{p_{min_i}}, y_{t_{min_i}}) \\
x_{right_i} &= \min(x_{p_{max_i}}, x_{t_{max_i}}) \\
y_{bottom_i} &= \min(y_{p_{max_i}}, y_{t_{max_i}}) \\
Intersection_i &= (x_{right_i} - x_{left_i}) \times (y_{bottom_i} - y_{top_i}) \\
Union_i &= Area_{p_i} + Area_{t_i} - Intersection_i \\
IoU_i &= \frac{Intersection_i}{Union_i}.
\end{aligned}
\tag{A4}
$$

For each pair of $\bar{B}_{p_i}$ and $\bar{B}_{t_i}$, the GIoU is:

$$
\begin{aligned}
x'_{left_i} &= \min(x_{p_{min_i}}, x_{t_{min_i}}) \\
y'_{top_i} &= \min(y_{p_{min_i}}, y_{t_{min_i}}) \\
x'_{right_i} &= \max(x_{p_{max_i}}, x_{t_{max_i}}) \\
y'_{bottom_i} &= \max(y_{p_{max_i}}, y_{t_{max_i}}) \\
Area'_i &= (x'_{right_i} - x'_{left_i}) \times (y'_{bottom_i} - y'_{top_i}) \\
GIoU_i &= IoU_i - \frac{Area'_i - Union_i}{Area'_i}.
\end{aligned}
\tag{A5}
$$

The GIoU loss for one pair of the predicted box and target box is:

$$
L_{GIoU_i} = 1 - GIoU_i.
\tag{A6}
$$

The GIoU loss for all pairs of predicted and target boxes is:

$$
L_{GIoU} = \frac{1}{n} \sum_{i=1}^{n} L_{GIoU_i}.
\tag{A7}
$$

## B.2 LOSSES FOR GESTURAL KEY POINTS

For the predicted eyes and fingertips or elbows and wrists, we use L1 loss. Specifically, each time, our model predicts m pairs of gestural key points. Each pair is denoted as $pair_{p_i} = (x_{eye_{p_i}}, y_{eye_{p_i}}, x_{fgt_{p_i}}, y_{fgt_{p_i}})$ (for VTL) or $pair_{p_i} = (x_{elb_{p_i}}, y_{elb_{p_i}}, x_{wst_{p_i}}, y_{wst_{p_i}})$ (for EWL). The ground truth gestural key points are represented as $pair_{t_i} = (x_{eye_{t_i}}, y_{eye_{t_i}}, x_{fgt_{t_i}}, y_{fgt_{t_i}})$ (for VTL) or $pair_{t_i} = (x_{elb_{t_i}}, y_{elb_{t_i}}, x_{wst_{t_i}}, y_{wst_{t_i}})$ (for EWL).

The loss for each predicted gestural pair is defined as:

$$
L_{gesture\_L1_i} = |x_{eye_{p_i}} - x_{eye_{t_i}}| + |y_{eye_{p_i}} - y_{eye_{t_i}}| + |x_{fgt_{p_i}} - x_{fgt_{t_i}}| + |y_{fgt_{p_i}} - y_{fgt_{t_i}}|
\tag{A8}
$$

for VTL, or

$$
L_{gesture\_L1_i} = |x_{elb_{p_i}} - x_{elb_{t_i}}| + |y_{elb_{p_i}} - y_{elb_{t_i}}| + |x_{wst_{p_i}} - x_{wst_{t_i}}| + |y_{wst_{p_i}} - y_{wst_{t_i}}|
\tag{A9}
$$

for EWL.

The L1 gestural key point loss is:

$$
L_{gesture\_L1} = \min L_{gesture_i}, i \in \{1, ..., m\}.
\tag{A10}
$$

Additionally, we apply a cross entropy loss $L_{gesture\_CE}$ for predicted gestural key points with two classes: "are gestural key points" and "are not gestural key points."

The total loss for gestural key points is:

$$L_{gesture} = \alpha_{gesture\_L1} \cdot L_{gesture\_L1} + \alpha_{gesture\_CE} \cdot L_{gesture\_CE}, \qquad (A11)$$

where $\alpha_{gesture\_L1} = 6$ and $L_{gesture\_CE} = 1.5$.

### B.3 SOFT TOKEN LOSS

For each object, we predict token spans produced by the BPE scheme (Sennrich et al., 2016), instead of object categories, and set the maximum number of tokens to 256 following Kamath et al. (2021). Following Kamath et al. (2021), we use a soft token loss $L_{token}$ for the predicted token spans. The soft token loss is a cross-entropy loss. Specifically, an object may correspond to k token locations ($1 \leqslant k \leqslant 256$), and the ground-truth probability for each of the k token locations is $\frac{1}{k}$.

### B.4 MATCHING STRATEGY

We match prediction and ground truth using the Hungarian algorithm (Kuhn, 1955; Carion et al., 2020) by minimizing the cost $C$:

$$C = \alpha_{L1} \cdot L_{L1} + \alpha_{GIoU} \cdot L_{GIoU} + \alpha_{token} \cdot L_{token}, \qquad (A12)$$

where $\alpha_{L1} = 5$, $\alpha_{GIoU} = 2$, and $\alpha_{token} = 1$.

### B.5 CONTRASTIVE ALIGNMENT LOSS

The contrastive alignment loss encourages alignment between the object feature from the Transformer decoder and the text feature from the Transformer encoder.

The number of predicted objects is $n$, and the number of text tokens is $l$. Let $F_{o_i}$ and $F_{t_j}$ denote the feature of the i th object and the feature of the j th token, respectively. At the same time, let $T_i^+$ denote the set of tokens to be aligned to by the i th object, and let $O_i^+$ denote the set of objects to be aligned to the i th token. Meanwhile, $\tau = 0.07$ is the temperature.

For all objects, the contrastive alignment loss is:

$$L_{contrastive_o} = \sum_{i=1}^{n} \frac{1}{|T_i^+|} \sum_{j \in T_i^+} -\log \frac{\exp(F_{o_i}^T F_{t_j}/\tau)}{\sum_{k=1}^{l} \exp(F_{o_i}^T F_{t_k}/\tau)}, \qquad (A13)$$

For all text tokens, the contrastive alignment loss is:

$$L_{contrastive_t} = \sum_{i=1}^{l} \frac{1}{|O_i^+|} \sum_{j \in O_i^+} -\log \frac{\exp(F_{t_i}^T F_{o_j}/\tau)}{\sum_{k=1}^{n} \exp(F_{t_i}^T F_{o_k}/\tau)}. \qquad (A14)$$

The final contrastive alignment loss is the average of $L_{contrastive_o}$ and $L_{contrastive_t}$:

$$L_{contrastive} = \frac{L_{contrastive_o} + L_{contrastive_t}}{2}. \qquad (A15)$$

## C COMPUTING COSINE SIMILARITIES USING DIFFERENT LINES

The cosine similarity in Equation (1) can be computed using different lines. Specifically, the eye, the fingertip, and the object can form a triangle, and there are three ways of choosing two lines from a triangle. In Equation (1), we use the two lines connected by the eye.

We investigate the effects of using different lines for cosine similarity computation. Specifically, we conducted an additional experiment using two lines connected by the fingertip. In other words, we use the following two vectors: one from the fingertip to the eye, and the other from the object center to the fingertip.

Our results (Table A1) show that using the lines connected by the eye and using the two lines connected by the fingertip can be regarded as fungible.

In specific, using the two lines connected by the fingertip, the model's performance is 69.0, 60.8, and 37.3 under the IoU threshold of 0.25, 0.50, and 0.75, respectively. Compared to the EWL model, it obtains a +1.5 performance boost under the IoU threshold of 0.75. Compared to the No Explicit Gestural Key Points Model, it obtains a +4.1 and +3.4 performance boost under the IoU threshold of 0.25 and 0.50, respectively.

Table A1: **Using different lines for cosine similarity computation in the VTL model.**

|                                          | IoU=.25 | IoU=.50 | IoU=.75 |
|------------------------------------------|---------|---------|---------|
| Ours (No explicit gestural key points)   | 64.9    | 57.4    | 37.2    |
| Ours (EWL)                               | **69.5** | 60.7   | 35.5    |
| Ours (VTL, cos_sim Vertex = Fingertip)   | 69.0    | **60.8** | **37.3** |

