# OpenReview forum: "Understanding Embodied Reference with Touch-Line Transformer"
_ICLR.cc/2023/Conference — ICLR 2023 poster_

### Official Review · Reviewer_qAaB · 2022-10-24

**Confidence:** 3
**Correctness:** 4
**Technical Novelty And Significance:** 3
**Empirical Novelty And Significance:** 2
**Recommendation:** 6

**Clarity, Quality, Novelty And Reproducibility:**

The work is introduced, explained, and validated well. It is of relatively high quality and clarity.

I believe the concept of VTL and its incorporation into a multi-modal transformer architecture is rather novel and original.

**Strength And Weaknesses:**

The introduction motivates the VTL well, and general approach is easy to understand, though the architecture section is somewhat brief.

While it is written well, the “Related Work” section is much too verbose and contains parts that are typically considered out-of-scope or tangential, in particular: “Gaze estimation”, “Saliency estimation”, and “HRI and collaboration”. Less relevant work could be removed to make the section more compact and to-the-point.

Overall, the results are quite impressive. It is clear that the VTL objective out-performs the use of EWL. The visual information provided by the imaged human is also useful (vs inpainting), and the proposed contribution overall out-performs recent works. It is also nice to see that the VTL-EWL discussion is further validated by seeing the cosine similarities defined in Eq. (1).

However, one must wonder whether there are additional ways to demonstrate the value of the contributed method. That is, are there other object-referring datasets that could be used where body keypoints are not annotated, but are visible and thus permit a VTL? I am not a domain expert so cannot make any recommendations, but the method and evaluation here seems far too specific and lack possibilities to generalize.

Miscellaneous:
- It is never really explained what the “(No Pose)” experiments are.

**Summary Of The Paper:**

This paper proposes a solution to “embodied reference understanding”, which aims to locate so-called “referents” which are objects referred to by text and gestures. The core observation of this work is that referents do not lie on the elbow-wrist line but rather on the eye-wrist line, which they call the “virtual touch line (VTL)”. By learning to predict the VTL using a multi-modal hybrid transformer architecture and by encouraging the co-linearity of object center, eye, and fingertip, the proposed method is shown to perform well on the YouRefIt dataset.

**Summary Of The Review:**

The paper was a good read, and the proposed solution is well packaged and delivered. However, the text could benefit from a mild revision, and the results on a single dataset might not necessarily warrant acceptance. Yet, I lean towards the positive side as the overall concept is simple, intuitive, and executed well.

**Post rebuttal comment.**
I thank the authors and the reviewers for their diligent comments. After having reviewed all available written materials, I feel that while the authors have made a good effort to respond to the various concerns, and their results do show meaningful improvements to the baseline of YouRefIt (Chen et al., 2021), it is difficult to enthusiastically recommend the paper's acceptance for the following reasons:
- the task is incredibly niche and this is self-evident by the lack of datasets to run evaluations on (and therefore show within-research-domain generality)
- no indications are made on how to extend the proposed concepts (hand-designed reference vector based on pre-trained keypoint network) to other areas of ML and CV (to show out-of-research-domain generality).

Given that the scope of NeurIPS is focused in the ML domain with allowance for competitive applications to CV, it is important for papers to provide knowledge that goes beyond a very niche CV topic. Yet, the paper is solid enough to be published at a top-tier CV+ML venue, so I retain my rating of borderline/weak accept.

---

> ### Author Response · Authors · 2022-11-14
> **Response to Reviewer qAaB**
>
> ## 3. Reviewer qAaB
>
> We thank R#iwkJ for constructive feedback, and we attempt to address raised issues one by one.
>
> ### 1.
> The introduction motivates the VTL well, and general approach is easy to understand, though the architecture section is somewhat brief.
>
> **Response:**
> We added a section named "A.2 ADDITIONAL LOSSES" in our appendix to particularize losses and matching strategy in our architecture. This section contains losses for predicted referent bounding boxes, losses for gestural key points, soft token loss, matching strategy, and contrastive alignment loss.
>
> ### 2.
> While it is written well, the “Related Work” section is much too verbose and contains parts that are typically considered out-of-scope or tangential, in particular: “Gaze estimation”, “Saliency estimation”, and “HRI and collaboration”. Less relevant work could be removed to make the section more compact and to-the-point.
>
> **Response:**
> We abridged our "Related Work" by removing the “Gaze estimation”, “Saliency estimation”, and “HRI and collaboration” sections.
>
> ### 3.
> However, one must wonder whether there are additional ways to demonstrate the value of the contributed method. That is, are there other object-referring datasets that could be used where body key points are not annotated, but are visible and thus permit a VTL? I am not a domain expert so cannot make any recommendations, but the method and evaluation here seems far too specific and lack possibilities to generalize.
>
> **Response:**
> Understanding the referred object using images with humans visible in them was proposed by the authors of YouRefIt [1] very recently. They call this task embodied reference understanding. They found that prior datasets were not suitable for this task because they were too small in size, did not have language or gestures, or were not natural enough. In other words, to the best of our knowledge, no other dataset is suitable for this task.
>
> However, as the authors of YouRefIt [1] argue, understanding the pointing gesture is of significant value. Our proposed virtual touch line exploits findings from psychology literature and significantly improves the computational model's performance in this direction.
>
> [1] Chen, Yixin, et al. "Yourefit: Embodied reference understanding with language and gesture." Proceedings of the IEEE/CVF International Conference on Computer Vision. 2021.
>
> ### 4.
> It is never really explained what the “(No Pose)” experiments are.
>
> **Response:**
> We revised the "IMPLICITLY LEARNED NONVERBAL SIGNALS" section of our paper to explain what the "(No Pose)" experiments are. We also changed "No Pose" to "No Explicit Gestural Key Points" in our tables to mitigate confusion.
>
> When conducting the "(No Pose)" experiments, we do not provide annotated gestural key points to our model or ask our model to predict these gestural key points. We still use the original input images in which humans are present. This experiment investigates whether our model can learn to use gestural information in images by itself without being explicitly asked to do so. Compared to the inpainting experiment, we find that our model could learn to use gestural information in the image by itself.

---

### Official Review · Reviewer_iwkJ · 2022-10-24

**Confidence:** 4
**Correctness:** 4
**Technical Novelty And Significance:** 3
**Empirical Novelty And Significance:** 3
**Recommendation:** 8

**Clarity, Quality, Novelty And Reproducibility:**

**Clarity**

What does the "no pose" ablation mean in Table 1 and how is it different from "inpainting"?


**Quality and Novelty**

The proposed method and implementation using the virtual touch-line are novel. The experimental results show the quality of the work.


**Reproducibility**

The authors have clearly explained all the components of their method, making it reproducible. They also provide their source code.

**Strength And Weaknesses:**

**Strengths**

1. The proposed design of the virtual touch-line is sound, intuitive, and well backed up by prior work.

2. The geometric consistency loss is also sound.

3. The ablation studies show the benefits of the individual components of the proposed end-to-end network.

4. The discussion of the limitations helps to understand the scope of the work.


**Weaknesses**

1. Looking at Eqn. 1, the authors take the reference line to be the one connecting the eye and the object. Have the authors considered the alternate, the line connecting the fingertip and the object? Is there any specific reason to choose one over the other, e.g., the noises in eye and fingertip detection?

2. I did not find an ablation study that uses only the gestural signals but no language signals. This could be an interesting ablation to observe, especially in the real-world context where the language input is noisy or corrupted.

**Summary Of The Paper:**

The authors present a method to automatically locate referents in scenes using a combination of embodied gesture signals and natural language descriptions. Their key contribution is the implementation of the "virtual touch-line", which is the extended line connecting the eye and the fingertip to localize objects in scenes. The authors propose a transformer-based network and a geometric consistency loss to combine the language and the gesture signals and predict the object bounding boxes. They demonstrate the benefits of their approach through experimental evaluations and ablation studies.

**Summary Of The Review:**

Overall, the paper solves a challenging problem using an intuitive and sound approach. The technical descriptions are clear and the experiments highlight the quality of the work. This method sets a new benchmark in embodied reference understanding and can be reused in associated applications.

---

> ### Author Response · Authors · 2022-11-14
> **Response to Reviewer iwkJ**
>
> ## 2. Reviewer iwkJ
> We thank R#iwkJ for constructive feedback, and we attempt to address raised issues one by one.
>
> ### 1.
> Looking at Eqn. 1, the authors take the reference line to be the one connecting the eye and the object. Have the authors considered the alternate, the line connecting the fingertip and the object? Is there any specific reason to choose one over the other, e.g., the noises in eye and fingertip detection?
>
> **Response:**
> Ideally, we want the three points--eye, fingertip, object--to be as co-linear as possible. To measure their co-linearity, we let these three points form two lines and measure how parallel the two lines are. We just want them to form two lines so that we can use the level of parallel of the two lines to measure the level of co-linearity of the three points. There is not any specific reason for us to let the fingertip and the eye form one line and let the object and the eye form another line. We believe the alternative way of forming one line using the eye and the fingertip and forming another line using the fingertip and the object would be equivalent. Additionally, there is a third way (there are three ways of choosing two lines from a triangle) of forming two lines: Let the object and eye form one line and let the object and fingertip form the other line. We believe the three ways of forming two lines are equivalent: The more co-linear the three points are, the more parallel the two lines are.
> We will conduct experiments to investigate this in the next few days if computational resources permit.
>
>
> ### 2.
> I did not find an ablation study that uses only the gestural signals but no language signals. This could be an interesting ablation to observe, especially in the real-world context where the language input is noisy or corrupted.
>
> **Response:**
> We are running this experiment. Results to be posted in the next few days
>
> ### 3.
> What does the "no pose" ablation mean in Table 1 and how is it different from "inpainting"?
>
> **Response:**
> We revised the "IMPLICITLY LEARNED NONVERBAL SIGNALS" section of our paper to explain what the “(No Pose)” experiments are. We also changed "No Pose" to "No Explicit Gestural Key Points" in our tables to mitigate confusion.
>
> When conducting the "(No Pose)" experiments, we do not provide annotated gestural key points to our model or ask our model to predict these gestural key points. We still use the original input images in which humans are present. This experiment investigates whether our model can learn to use gestural information in images by itself without being explicitly asked to do so. Compared to the inpainting experiment, we find that our model could learn to use gestural information in the image by itself.
>
> The differences between the "No Pose" experiment and the "Inpainting" experiment lie in the input images. We use original images in which humans are present for the "No Pose" experiment. We use inpainted images in which humans are removed for the "Inpainting" experiment.

---

> ### Author Response · Authors · 2022-11-17
> **Experiment Results when using Corrupted Language Inputs**
>
> We completed **the ablation study that uses only the gestural signals but no language signals**.
>
> Specifically, we corrupt language inputs by changing each input sentence to the word "object." After re-training our models, we observe more than 35% performance drops under the IoU threshold of 0.75 both when explicitly (VTL) and non-explicitly (No Explicit Gestural Key Points) asking our model to learn gestural keypoints. Without re-training, corrupted language inputs result in a 79% and a 66% performance drop under the IoU threshold of 0.75 to the **VTL** model and to the **No Explicit Gestural Key Points** model, respectively. We added a table to our revised paper to show the results of this set of experiments.
>
> **With re-training**, the performance of the **VTL** model under the IoU threshold of 0.25, 0.50, and 0.75 is **49.4, 41.7, and 23.9**, respectively; the performance of the **No Explicit Gestural Key Points** model under the IoU threshold of 0.25, 0.50, and 0.75 is **49.2, 42.1, and 23.4**, respectively.
>
> **Without re-training**,   the performance of the **VTL** model under the IoU threshold of 0.25, 0.50, and 0.75 is **31.4, 25.3, and 13.1**, respectively; the performance of the **No Explicit Gestural Key Points** model under the IoU threshold of 0.25, 0.50, and 0.75 is **19.5, 16.0, and 7.8**, respectively.

---

> ### Author Response · Authors · 2022-11-18
> **Experiment Results when Computing Cosine Similarity using the Line Connecting the Fingertip and the Object**
>
> "Looking at Eqn. 1, the authors take the reference line to be the one connecting the eye and the object. Have the authors considered the alternate, the line connecting the fingertip and the object? Is there any specific reason to choose one over the other, e.g., the noises in eye and fingertip detection?"
>
>
> **Response:**
> We conducted additional experiments using the alternate mentioned above. Our results show that the alternate and our way in the paper **can be regarded as fungible**.
>
> In specific, using the alternate, the model's performance is 69.0, 60.8, and 37.3 under the IoU threshold of 0.25, 0.50, and 0.75, respectively. Compared to the **EWL model**, the **VTL model using the alternate** obtains a +1.5 performance boost under the IoU threshold of 0.75. Compared to the **No Explicit Gestural Key Points Model**, the **VTL model using the alternate** obtains a +4.1 and +3.4 performance boost under the IoU threshold of 0.25 and 0.50, respectively.
>
> We added a section (A.3) to our appendix to include these results.

---

> ### Comment · Reviewer_iwkJ · 2022-11-22
> **Reviewer Response**
>
> Thank you for the detailed response. I do not have further questions and maintain my original recommendation.

---

### Official Review · Reviewer_BH2V · 2022-10-24

**Confidence:** 3
**Correctness:** 4
**Technical Novelty And Significance:** 3
**Empirical Novelty And Significance:** 3
**Recommendation:** 6

**Clarity, Quality, Novelty And Reproducibility:**

Overall, this work has a few novel ideas (e.g., virtual touch line, using pose) and makes a significant improvement in performance on embodied reference understanding. The writing is overall clear and the results are strong. However, it is not fully clear where the performance increase comes from so the paper would benefit from a more in depth discussion and analysis of the model and results.

**Strength And Weaknesses:**

Overall, I think this is an interesting paper that shows a few modifications to a multi-modal transformer based setup leads to a significant increase in performance in embodied reference understanding. Note that I'm not familiar with this literature very well (embodied reference understanding seems to be a problem recently introduced), so my assessment of novelty/originality might not be 100% accurate. As far as I understand, the main contributions here are the multi-modal transformer and the predicting virtual touch line alongside the bounding box of referent. I like the virtual touch line idea (and the argument/results for why you would prefer it to EWL) and this seems to increase performance. Overall, the results are pretty strong but it is not fully clear where this increase in performance comes from (more on this below). I'd say the writing is overall pretty clear but some model and evaluation details are missing, which makes it difficult to judge the full merit of the approach (again more on this below).

- What exactly leads to better performance from this model? Or to put it another way, what makes this model work much better than the Chen et al.'s model? Looking at the results, even without any pose information or humans in the image, the method in this paper does much better than Chen et al's? Why is this the case? I think it'd be great to have an ablation analysis that shows where this increase in performance comes from.
- The requirement of pose annotation (and VTL) limits the applicability of the approach slightly, but the model seems to do better than previous ones even without any pose information. So it'd be nice to know why.
- An alternative could be to use a pre-trained pose estimation network and use the features from this model as additional inputs. Have the authors tried this?
- What parts, if any, are pre-trained in this model? It looks like the model builds on a previous object detection model. Is this pre-trained? This has to be mentioned in the paper.
- None of the results tables have standard deviations. Even though the performance gap between models is pretty large, it'd be nice to have these std devs.
- In Table 2, when you train with EWL, do you still have the geometric consistency loss? If so, wouldn't this hurt performance since EWL usually doesn't lie on a line with the referent.
- What happens if you don't have the geometric consistency loss (but still predict VTL)? How much does this hurt performance?


**Summary Of The Paper:**

This paper presents a transformer based technique for embodied reference understanding. In this problem, the task is to determine the object being referred to (by the human in the image and the text) in an image. The model takes in visual and text features (from a CNN and BERT respectively) and outputs 1) the bounding box of the referent (the object being referred to) 2) touch-line, the vector from the human (referrer) to the referent (referred object). The motivation for including the touch line is that this should help the model locate the referred object. The model is trained to predict both of these outputs. For the touch-line, the authors annotate the datasets themselves to provide this information. They try two different ways to define the touch-line. One is the vector from elbow to wrist (EWL), and the other is the one from the eye to fingertip (virtual touch line, VTL). Citing studies from psychology, the authors argue that VTL is a better choice and in fact show that predicting VTL leads to better referent detection than using EWL. The model uses an additional geometric loss to encourage the predicted touch line and the predicted referred object to lie on the same line. Overall, their model is able to reach SoTA performance on the YouRefIt embodied reference understanding dataset.

**Summary Of The Review:**

Given the significant increase in performance on the embodied reference understanding task and the novel idea of coupling virtual touch line prediction with referent detection, I think this would be a good addition to the conference. However, I'd have liked to see a more detailed analysis of the performance of the model.

---

> ### Author Response · Authors · 2022-11-14
> **Response to Reviewer BH2V (first half)**
>
> ## 1. Reviewer BH2V:
>
> We thank R#BH2V for constructive feedbacks, and we attempt to address raised issues one by one.
>
> ### 0.
> They try two different ways to define the touch-line. One is the vector from elbow to wrist (EWL), and the other is the one from the eye to fingertip (virtual touch line, VTL). Citing studies from psychology, the authors argue that VTL is a better choice and in fact show that predicting VTL leads to better referent detection than using EWL.
>
> **Response:**
> We would like to clarify that we define the VTL as the line that connects the eye and the fingertip. Researchers in the psychology study [1] named the line touch line because they believe that "pointing originates in touch" and the touch line, from the point of view of the person who is initiating the pointing gesture, would indicate the object the person is trying to touch. The line that connects the elbow and the wrist, by our definition, is not a touch line.
>
> [1] Cathal O’Madagain, Gregor Kachel, and Brent Strickland. The origin of pointing: Evidence for the touch hypothesis. Science Advances, 5(7):eaav2558, 2019. 2
>
>
> ### 1.
> Overall, the results are pretty strong but it is not fully clear where this increase in performance comes from (more on this below). What exactly leads to better performance from this model? Or to put it another way, what makes this model work much better than the Chen et al.'s model? Looking at the results, even without any pose information or humans in the image, the method in this paper does much better than Chen et al's? Why is this the case? I think it'd be great to have an ablation analysis that shows where this increase in performance comes from. The requirement of pose annotation (and VTL) limits the applicability of the approach slightly, but the model seems to do better than previous ones even without any pose information. So it'd be nice to know why.
>
> **Response:**
> As we stated in the paper title, the abstract, and the intro, the increase in performance was resulted from the combination of the pre-trained MDETR and our proposed **virtual touch line** and geometric consistency loss. The pre-trained MDETR model, which had a strong capacity for multi-modal tasks, could account for part of the performance increase. Additionally, our proposed virtual touch line (or the elbow-wrist line) and losses could account for the rest of the performance increase.
>
> ### 2.
> An alternative could be to use a pre-trained pose estimation network and use the features from this model as additional inputs. Have the authors tried this?
>
> **Response:**
> We have considered using a pre-trained pose estimation network such as PAF [1]. However, we notice that PAF, similar to the de facto human pose representation in modern computer vision defined by COCO [2], does not explicitly incorporate eyes and fingertips. **We have already stated this problem on the bottom 3 lines of the first page and the top 4 lines of the second page.**
>
> [1] Cao, Zhe, et al. "Realtime multi-person 2d pose estimation using part affinity fields." Proceedings of the IEEE conference on computer vision and pattern recognition. 2017.
>
> [2] Tsung-Yi Lin, Michael Maire, Serge Belongie, James Hays, Pietro Perona, Deva Ramanan, Piotr Dollár, and C Lawrence Zitnick. Microsoft coco: Common objects in context. In European Conference on Computer Vision (ECCV), 2014. 2
>
> ### 3.
> What parts, if any, are pre-trained in this model? It looks like the model builds on a previous object detection model. Is this pre-trained? This has to be mentioned in the paper.
>
> **Response:**
> The image encoder and the text encoder are pre-trained-ResNet and pre-trained BERT, respectively. Additionally, the transformer decoder is a pre-trained MDETR decoder. We added this information to our revised paper.

---

> ### Author Response · Authors · 2022-11-14
> **Response to Reviewer BH2V (second half)**
>
> ### 4.
> None of the results tables have standard deviations. Even though the performance gap between models is pretty large, it'd be nice to have these std devs.
>
> **Response:**
> Thanks for pointing that out. We will try our best to run experiments and compute the std before the rebuttal deadline (November 18.) However, due to limited computational resources, we won't be able to guarantee that we can catch the rebuttal deadline.
>
> ### 5.
> In Table 2, when you train with EWL, do you still have the geometric consistency loss? If so, wouldn't this hurt performance since EWL usually doesn't lie on a line with the referent.
>
> **Response:**
> We do have the geometric consistency loss when training our model with EWLs. Having this loss would help our model locate the rough location of the referents. Specifically, while referents do not precisely lie on EWLs, EWLs can still be used to provide a rough direction of the referent. Our experiment results in Table 2 corroborate this: The model's ability in locating the rough locations of the referents (IoU=0.25 and IoU=0.75) is improved (by 4.6% and 3.3%, respectively) when trained with EWls and the geometric consistency loss.
>
> In contrast, since referents usually do not lie very precisely on EWLs, EWLs and the geometric consistency loss did cause detriments to the model's capacity in very accurately locating referents (a 1.7% performance drop under the IoU threshold of 0.75).
>
> ### 6.
> What happens if you don't have the geometric consistency loss (but still predict VTL)? How much does this hurt performance?
>
> **Response:**
> In Table 5, we compared the model performance when having and when not having the geometric consistency loss. In our revised paper, we clarify that "referent alignment loss" is another name for "geometric consistency loss." The results in Table 5 show that, if we don't have the geometric consistency loss (but still predict VTL), the performance is impaired (by 4.0%, 4.5%, and 2.6% under the IoU threshold of 0.25, 0.50 and 0.75, respectively.)

---

> ### Author Response · Authors · 2022-11-18
> **Standard Deviations**
>
> Thank you again for pointing this out. However, due to limited computational resources, we are unable to provide std devs before the rebuttal deadline.

---

### Decision · Program_Chairs · 2023-01-20

**Decision:**

Accept: poster

**Justification For Why Not Higher Score:**

The main reason to accept the paper as a poster instead of as a spotlight is the fact that the paper does not propose a technique that can be applied to different problems or domains. As pointed out implicitly or explicitly by the reviewers, this paper addresses a very specific Computer Vision problem and proposes a very interesting and tailored solution. Thus, the ICLR audience interested in this work might be not as broad as for more general ML approaches.

**Justification For Why Not Lower Score:**

As discussed in the metareviews summary, all the reviewers consider that the paper has significant contributions, and they agree on accepting the paper.

**Metareview: Summary, Strengths And Weaknesses:**

This paper addresses the problem of embodied reference understanding (locating referents in an image using embodied gestural cues and language references). The main idea is the design of the touch-line transformer, which takes tokenized visual and textual features to simultaneously predict the referent’s location and the touch-line vector. Additionally, a geometric consistency loss that promotes the co-linearity between referents and touch lines is applied. The experiments are conducted on the YouRefIt dataset.

The main strengths of this paper are the interest of the main idea (which takes inspiration from psychology studies that show the relevance of the virtual touch-line -considered in this work and ignored by previous works); the obtained results (which significantly outperform previous methods on YouRefIt dataset); and the ablation studies, that show the contribution of each component and help understanding the results. Some concerns of the reviewers (e.g. need for additional details in the evaluation) were addressed by the authors in their feedback.

Overall, this paper proposes an interesting new technical solution to the problem of embodied reference understanding, which is an important problem in HRI and HR collaboration.

**Note From Pc:**

if the above contains the word "oral" or "spotlight" please see: "oral" presentation means -> notable-top-5% and "spotlight" means -> notable-top-25%. As stated in our emails, we are disassociating presentation type from AC recommendations